# Blood–Brain Barrier Impairment in Patients Living with HIV: Predictors and Associated Biomarkers

**DOI:** 10.3390/diagnostics11050867

**Published:** 2021-05-12

**Authors:** Giulia Caligaris, Mattia Trunfio, Valeria Ghisetti, Jessica Cusato, Marco Nigra, Cristiana Atzori, Daniele Imperiale, Stefano Bonora, Giovanni Di Perri, Andrea Calcagno

**Affiliations:** 1Department of Medical Sciences, Faculty of Medicine and Surgery, University of Torino, 10126 Torino, Italy; mattia.trunfio@edu.unito.it (M.T.); jessica.cusato@unito.it (J.C.); daniele.imperiale@aslcittaditorino.it (D.I.); stefano.bonora@unito.it (S.B.); giovanni.diperri@unito.it (G.D.P.); andrea.calcagno@unito.it (A.C.); 2Laboratory of Microbiology and Molecular Biology, Ospedale Amedeo di Savoia, ASL TO2, 10149 Torino, Italy; valeria.ghisetti@gmail.com; 3Diagnostic Laboratory Unit, San Giovanni Bosco Hospital, 10154 Torino, Italy; marco.nigra@aslcittaditorino.it; 4Unit of Neurology, Maria Vittoria Hospital, ASL Città di Torino, 10144 Torino, Italy; domp@aslcittaditorino.it

**Keywords:** blood–brain barrier, biomarkers, HIV

## Abstract

Despite the substantial changes resulting from the introduction of combination antiretroviral therapy (cART), the prevalence of HIV-associated neurocognitive disorders (HAND) remains substantial. Blood–brain barrier impairment (BBBi) is a frequent feature in people living with HIV (PLWH) and it may persist despite effective antiretroviral treatment. A cross-sectional study was performed in PLWH who underwent lumbar puncture for clinical reasons or research protocols and several cerebrospinal fluid biomarkers were studied. BBBi was defined as cerebrospinal fluid-to-serum albumin ratio (CSAR) >6.5 (<40 years) or >8 (>40 years). We included 464 participants: 147 cART-naïve and 317 on cART. Male sex was prevalent in both groups (72.1% and 72.2% respectively); median age was 44 (38–52) years in naïve and 49 (43–57) years in treated subjects. BBBi was observed in 35.4% naïve and in 22.7% treated participants; the use of integrase inhibitors was associated with a lower prevalence (18.3 vs. 30.9%, *p* = 0.050). At multivariate binary logistic regression (including age and sex) nadir CD4 cell count (*p* = 0.034), presence of central nervous system (CNS) opportunistic infections (*p* = 0.024) and cerebrospinal fluid (CSF) HIV RNA (*p* = 0.002) in naïve participants and male sex (*p* = 0.021), a history of CNS opportunistic infections (*p* = 0.001) and CSF HIV RNA (*p* = 0.034) in treated patients were independently associated with BBBi. CSF cells and neopterin were significantly higher in participants with BBBi. BBBi was prevalent in naïve and treated PLWH and it was associated with CSF HIV RNA and neopterin. Systemic control of viral replication seems to be essential for BBB integrity while sex and treatment influence need further studies.

## 1. Introduction

There are approximately 38 million people living with HIV worldwide [1], out of which 20–50% are estimated to develop a certain degree of cognitive impairment [2]. Despite combination antiretroviral therapy (cART) that has substantially contributed to reducing the HIV-related complications and mortality, HIV-associated neurocognitive disorders (HAND) continue to be relevant, especially considering their prevalence on a global scale [3]. Blood–brain barrier impairment (BBBi) may play a crucial role in the pathogenesis of HAND. In this regard, up to 22% of asymptomatic HIV-positive subjects, 50% of patients suffering from AIDS and up to 100% of patients with HIV-associated dementia (HAD) showed an increased BBB permeability [4].

Monocyte transit across the BBB is a pivotal process in HIV central nervous system (CNS) infection [5] and several mechanisms have been involved in the pathogenesis of HAND, including neuroinflammation, antiretroviral neurotoxicity, tight junction dysregulation and the role of Tat and gp120 as neurotoxic viral proteins able to induce dose-dependent oxidative stress directly damaging BBB integrity [6]. Moreover, since co-receptors that could be used by HIV to enter CD4+ T cells have been detected in human astrocytes, a key cell in maintaining BBB integrity [6], several studies support the idea that astrocytes play a key role in the pathogenesis of HAND and they should be considered as a target for therapy [7,8,9]. Indeed, it is evident that BBB impairment is associated with vascular damage, permeability alteration and the accumulation of toxins [6].

A number of neurodegenerative disorders, such as Alzheimer’s disease (AD), are associated with BBBi, microvascular and neurovascular degeneration [10]. Moreover, even though vascular dysfunction in AD has been usually ascribed to the accumulation of beta--amyloid and tau [11], a recent study suggests that neurovascular dysfunction associated with BBBi is an early biomarker of cognitive decline in AD regardless of beta-amyloid and tau abnormalities [12].

BBBi has been observed despite cART and it has been associated with neuronal damage biomarkers, such as increased levels of total tau and phosphorylated tau [7]. Several blood and cerebrospinal fluid (CSF) biomarkers and imaging investigations have been examined as potential HAND biomarkers, yet so far none of them has proved sufficient accuracy in diagnosing and monitoring patients affected by HAND [13].

The aim of this study was to describe demographic, clinical and therapeutic characteristics, risk factors, comorbidities, as well as CSF and plasma biomarkers in both naïve and treated PLWH, potentially associated with BBBi.

## 2. Materials and Methods

A cross-sectional study was performed involving patients with confirmed HIV infection, who underwent lumbar puncture for clinical reasons or research protocols and who were capable to sign the informed consent. The trial was approved by Ethics Committee of the University of Turin (prospective study on predictors of neurocognitive decline in HIV-positive patients PRODIN, protocol code 103/2015 approved on 22 June 2015).

Demographic data, risk factors, co-infections, psychiatric comorbidities, therapeutic and immunovirological data were recorded. The cerebrospinal fluid-to-serum albumin ratio (CSAR), calculated as CSF albumin (mg/L)/serum albumin (g/L), was used to evaluate BBB integrity.

Blood–brain barrier damage definition was derived from age-adjusted Reibergrams (normal if below 6.5 in patients aged <40 years and below 8 in patients >40 years) [14].

CSF total tau (t–tau), phosphorylated tau (p–tau), and beta–amyloid1–42 (Abeta1–42) were measured by immunoenzymatic methods (Innogenetics, Ghent, Belgium, EU) with limits of detection, respectively, of 87, 15, and 87 pg/mL. Neopterin was measured through validated ELISA methods (DRG Diagnostics, Marnurg, Germany, EU). Reference values were as follows: t–tau < 300 pg/mL (in patients aged 21–50), <450 pg/mL (in patients aged 51–70), and <500 pg/mL in older patients; p–tau < 61 pg/mL; 1–42 beta-amyloid > 500 pg/mL; neopterin < 1.5 ng/mL HIV–RNA was quantified by the Roche Amplicor assay v2.0 (Hoffman–La Roche, Basel, Switzerland) with a lower limit of quantification of 20 copies/mL.

HAND was diagnosed according to the Frascati criteria [15]; the neurocognitive evaluation was based on 14 tests, assessing eight different cognitive domains; patients were categorized as having either asymptomatic (ANI) or mild neurocognitive impairment (MND), or HIV-associated dementia (HAD).

Data were analyzed using nonparametric statistical methods: variables were described with medians (interquartile ranges, IQR), absolute values (proportion) or ranges (minimum–maximum). Then, the associations between these variables with BBBi were assessed using Spearman’s test for continuous numerical variables, Mann–Whitney and Fisher’s exact/Chi^2^ test for categorical ones. A multivariate analysis was performed to select the determinants independently associated with BBBi: we used a binary logistic regression analysis using variables with a *p*-value <0.05 at bivariate comparisons. Data analysis was performed using SPSS software for Mac (version 22.0, IBM Corp, Armonk, NY, USA).

## 3. Results

### 3.1. Participants’ Demographic and Clinical Features

Four hundred sixty-four patients were included, of which 147 were cART-naïve (median age of 44 years) and 317 were cART-treated (median age of 49 years); their demographic features, risk factors and comorbidities are shown in Table 1.

### 3.2. Antiretroviral Naïve and Treated Participants

Both groups of patients consisted mostly of male subjects (72.1% of naïve and 72.2% of ART-treated subjects). With regard to the co-infections, HCV, past syphilis and toxoplasmosis were prevalent among ART-treated subjects (HCV-positive 28.2% vs. 16.8%; past syphilis 22.0% vs. 21.4% and toxoplasmosis 42% vs. 31.2% in cART-treated and cART naïve, respectively).

Clinical and cART-related features of the study participants are shown in Table 2. Clinical categories were represented by: HAND (asymptomatic or mild neurocognitive impairment, and HIV-associated dementia), primary HIV acute infections, late presenters, CNS opportunistic infections, HIV–encephalitis, CSF viral escape or rebound encephalitis, isolated white matter hyperintensities, leukoencephalopathy, syphilis or neurosyphilis, and other CNS disorders. HAND prevalence was 28.5% among treated subjects versus 10.1% in naïve patients.

Plasma and CSF characteristics and bivariate comparisons among naïve and treated subjects are shown in Table 3. Median CD4+ count nadir was higher in ART-treated patients (97 cell/mm^3^ versus 49 cell/mm^3^ of naïve patients), as well as current CD4+ T cell count (366 cell/mm^3^ in ART-treated subjects versus 57 CD4+/mm^3^ in naïve). With regard to CSF biomarkers, naïve subjects had higher neopterin (2.96 vs. 0.94 ng/mL, *p* <0.001), but lower tau (165 vs. 222 pg/mL, *p* = 0.045) and p–tau levels (33 vs. 37 pg/mL, *p* = 0.040); Beta42 (962 vs. 919 pg/mL, *p* = 0.901) and S100Beta (145 vs. 129 pg/mL, *p* = 0.758) were similar between the two groups.

### 3.3. Blood–Brain Barrier Impairment and CSF Biomarkers

BBBi was observed in 35.4% of naïve patients and in 22.7% of ART-treated people; clinical, demographic, and therapeutic variables stratified according to treatment group and presence/absence of BBBi are shown in Table 4 and depicted in Figure 1.

Given the potentially different mechanisms underlying BBBi in naïve and cART receiving participants, we stratified analysis on predictors and biomarkers according to treatment status.

In naïve participants we observed that BBBi was associated with younger age (41 vs. 46 years, *p* = 0.034), higher CSF HIV RNA (4.36 vs. 3.71 Log_10_ copies/mL, *p* = 0.002) and with the presence of CNS opportunistic infections (25 vs. 6.3%, *p* <0.002). In treated participants BBBi was associated with male sex (30.6 vs. 18.1%, *p* = 0.037), higher CSF HIV RNA (1.53 vs. <1.28 Log_10_ copies/mL, *p* = 0.029), a history of CNS opportunistic infections (22.2 vs. 7.3%, *p* <0.001) and with non INSTI based regimens (30.9 vs. 18.3%, *p* = 0.050).

Of note, demographic, clinical and immunovirological features were not statistically different among INSTI and other-ARV recipients with the exception of a longer time since first positive HIV serology (167 vs. 124 months, *p* = 0.046) in INSTI–receivers.

JCV, CMV and EBV DNA were detected more commonly in participants with BBBi with statistically significant differences for CMV DNA (in naïve subjects) and EBV DNA (in treated individuals) (Figure 2).

Besides higher CSF HIV RNA, we observed significantly higher levels of CSF neopterin in participants with BBBi (Figure 3).

At multivariate binary logistic regression (including age and sex) we identified nadir CD4 cell count (*p* = 0.034, for 100 cells/uL increase aOR 1.401, 95% CI 1.026–1.912), presence of CNS opportunistic infections (*p* = 0.024, aOR 4.193, 95% CI 1.207–14.565) and CSF HIV RNA (*p* = 0.002, aOR for 1 Log10 increase 1.798, 95% CI 1.245–2.595) in naïve participants.

Aside from the aforementioned factors, we included the use of INSTI in the multivariate model for cART-treated participants: male sex (*p* = 0.021, aOR 3.230, 95% CI 1.191–8.755), a history of CNS opportunistic infections (*p* = 0.001, aOR 5.439, 95% CI 2.054–14.405) and CSF HIV RNA (*p* = 0.034, aOR for 1 Log_10_ increase 1.336, 95% CI 1.022–1.747) were independently associated with BBBi.

## 4. Discussion

We studied the prevalence of BBBi and a large set of variables in order to identify what may predict this event. We observed a prevalence of BBB impairment of 35.4% in ART-naïve and of 22.7% in cART-treated PLWH supporting the evidence that BBB alterations may persist despite antiretroviral therapy.

We have also identified female sex and cART therapy as independent protective factors for BBBi. In particular, male participants showed a higher prevalence of BBB alteration, both in ART-treated and naïve subjects. Furthermore at multivariate analysis female sex was an independent predictor of BBB integrity.

This difference could be ascribed to a greater prevalence of risk factors in males, such as hepatitis B (HBV), whose rate of infection is higher in men than in women in all Mediterranean countries [16]. Furthermore, male sex plays a key role in the progression of severe forms of chronic liver diseases [17].

It is worth noting that a major problem of the literature is the clear inequality of female representation in the study populations. Women constitute 51% of HIV-positive subjects worldwide and sex differences in HIV infection have been highlighted in several studies conducted both before and after the introduction of cART [18]. Nevertheless, several studies investigate sex differences and the prevalence of neurocognitive disorders in the female population, but the conclusions are not unequivocal. Some authors, as Maki and Martin–Thormeyer, highlight that there could be a higher risk of cognitive disorder in HIV-infected women [19]. Other studies reached similar conclusions, in particular suggesting that this difference is higher in some cognitive domains, such as memory and learning [20]. By contrast, in a recent study, Namagga and colleagues show higher prevalence and risk factors of HAND in men compared with women [21]. Finally, another evidence indicates that there are no substantial differences in cognitive impairment based on sex, except a higher risk of cognitive deficiency in the psychomotor domain for women [22]. These conflicting results suggest that further studies that specifically investigate the female population are needed especially considering the sex differences concerning the immune response, the pathogenesis of HIV infection, the pharmacological and pharmacodynamic responses [18], as well as the alteration of the blood–brain barrier [23].

In this study cART therapy resulted as an independent protective factor of BBBi. Interestingly, the effect of antiretroviral therapy on BBB has not been clearly demonstrated in literature despite similar analytical methods. Indeed, in their study, Abdulle and colleagues did not find a significant correlation between cART treatment and the reduction of BBB permeability [24]. Furthermore, a recent study highlights that despite CSF and plasma HIV–RNA suppression, neurofilament light chain (NFL) level and albumin ratio did not change after cART therapy [25]. Among the studied biomarkers, we observed a significant reduction in CSF 179 neopterin (suggesting the reduction in CNS inflammation and immune activation) and no change in 42–beta amyloid and S–100 prompting towards no effect of amyloid metabolism and astrocytosis. Interestingly, we noted higher total tau and phosphorylated tau in treated individuals suggesting either a detrimental effect of treatment on axonal integrity or the potential for age-associated damage potentially in common with Alzheimer’s dementia [26]. Despite years of research, a clear benefit of specific ARVs or combinations in improving CNS virological control and patients’ neurocognitive performance is unclear.

Our univariate analysis shows a lower prevalence of BBB alteration among participants treated with integrase inhibitors (INSTI), compared with patients treated with other classes of antiviral drugs (protease inhibitors or non–nucleoside reverse transcriptase inhibitors). Integrase inhibitors (Raltegravir, Elvitegravir, Dolutegravir and Bictegravir) are commonly used as first-line cART regimen, in association with other classes of antivirals, such as two nucleoside reverse transcriptase inhibitors [27]. Moreover, other elements can influence the therapy response, such as HIV DNA, substance abuse, pharmacogenetics, adherence to antiretrovirals and demographic factors, e.g., age and ethnicity [28]. Considering that, the current study shows two limitations worth mentioning: firstly, in the multivariate analysis not all of these factors have been considered as they were not available; secondly, a wider range of patients was treated with INSTI compared to those treated with other classes of drugs (28%, versus 17.4% NNRTI treated patients, 12.9% treated with PI and 5% with Maraviroc).

The association found between BBBi and CSF viral load, both among ART-naïve and treated patients, confirms the evidence observing an increase in CSF HIV–RNA among patients affected by neurological symptoms which were associated with BBBi [28]: a direct or indirect pathogenic mechanism linking uncontrolled HIV replication and BBB integrity seem reasonable. A persistent compartmental HIV replication (and potentially of other persistent viruses) may increase cell trafficking and immune activation. Inflammation seems to be a key factor in influencing BBB function and integrity since pleocytosis and abnormally high CSF neopterin were more prevalent in participants with BBBi. Although in recent studies neopterin was not associated with BBBi [7], the results of the current analysis support this concept. Indeed, a recent study has demonstrated that neopterin levels are elevated in CSF of untreated HIV patients, then its concentrations decrease as cART is started, and eventually reaches a plateau persisting at higher levels despite ART-therapy compared to HIV-negative controls [29].

In our analysis, EBV and CMV infections were more common among ART-naïve patients. The incidence of CMV infection has significantly decreased with cART introduction [30], despite that in their study Perello et al. [31] reported a higher incidence of acute CMV infection in the last years than in the early period of cART. In this regard, as in pre-cART era CMV coinfection caused a major risk of progression to AIDS, several studies show that even in cART epoch the presence of CMV in the blood is associated with a worse prognosis, cause of the increased risk of CMV disease progression, AIDS-defining diagnosis and death [32]. As reported in literature, our results confirm the less prevalence of CMV infection in cART treated patients, although CMV remains a relevant comorbidity even in the cART era.

EBV infects more than 90% of the worldwide population and it is associated with several cancers in PLWH, including Hodgkin lymphomas, non-Hodgkin lymphomas and Burkitt lymphomas [33]. Dehee and colleagues found a similar rate of EBV detection among 227 HIV-positive subjects and controls [34], but higher viral loads in the former ones: despite EBV replication may just represent a less controlling immune system studying EBV may be useful for understanding chronic immune activation and some HIV-associated non-infectious comorbidities [35].

Finally, the mechanism that underlies BBB impairment and HAND is not totally clear. HAND could be a direct consequence of the BBB breakdown by different mechanisms, such as an increased permeability, that allow an increasing entry of virus to CNS [36], or a higher neurotoxicity, caused by an increased drug concentration [37]; furthermore BBB disruption may reflect astrocytes and neurons alteration [8,28]. This is beyond the aim of the current study since we did not specifically analyze the link between BBB integrity and neurocognition.

## 5. Conclusions

BBB impairment is a common finding in PLWH affected by neurological or neurocognitive disorders either with or without suppressive cART. Indeed, BBB plays a critical role in CNS physiology and in neurological disorders, such as neurodegenerative disorders.

We observed higher levels of HIV RNA and neopterin in the CSF of study participants with BBBi, highlighting the significant role of inflammation in the pathogenesis of BBBi. Aside from virological control, female sex was found to be protective: further studies are needed to confirm this finding and to understand thelong–term consequences of an impaired BBB in PLWH.

## Figures and Tables

**Figure 1 diagnostics-11-00867-f001:**
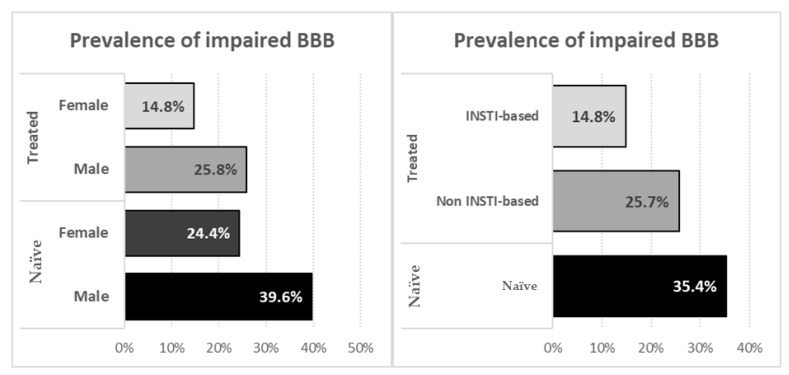
Prevalence of impaired blood–brain barrier stratified according to sex (**left** panel) and to treatments (**right** panel).

**Figure 2 diagnostics-11-00867-f002:**
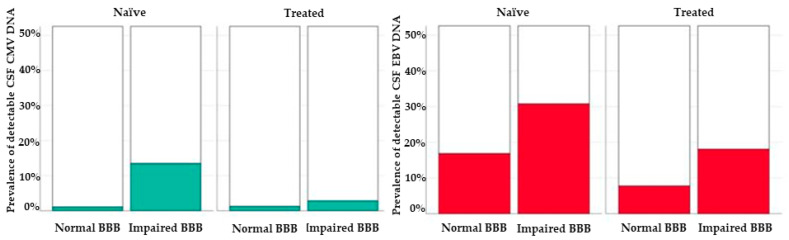
Prevalence of detectable CMV (**left**, green bars) and EBV (**right**, red bars). DNA in the cerebrospinal fluid of study participants according to blood–brain barrier integrity and treatment status.

**Figure 3 diagnostics-11-00867-f003:**
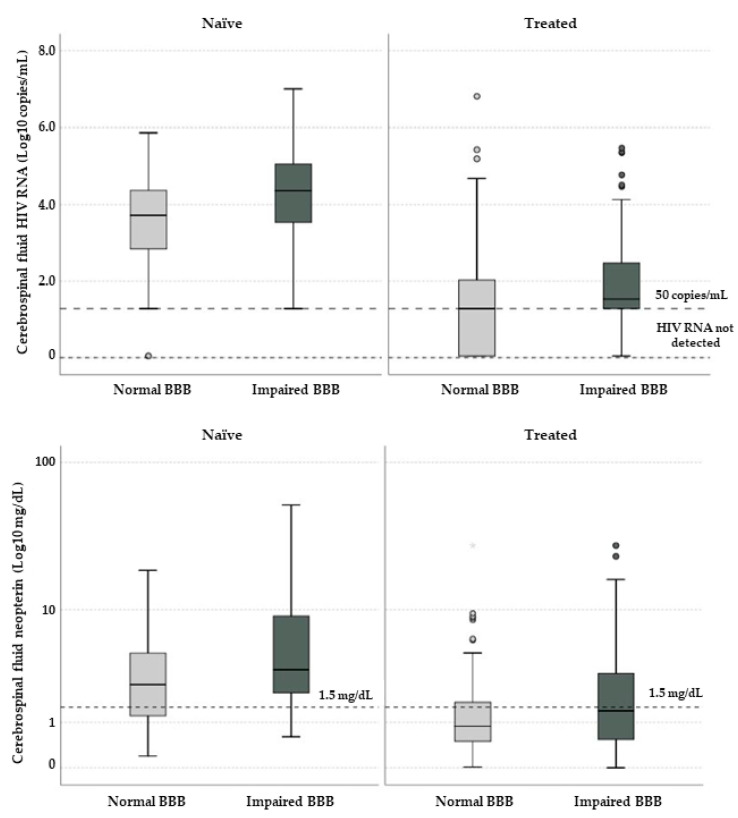
Cerebrospinal fluid HIV RNA (**above**) and neopterin (**below**) in study participants according to blood–brain barrier integrity and treatment status. Horizontal lines and boxes represent median values and interquartile ranges; whiskers show 10th and 90th percentiles while circles and stars are outliers and extreme outliers. In the graph above dotted horizontal lines represent 50 copies/mL and “target not detected” values; in the one below the horizontal dotted line represents the proposed threshold for cerebrospinal fluid neopterin (1.5 mg/dL).

**Table 1 diagnostics-11-00867-t001:** Demographic features, risk factors and comorbidities of study participants.

Characteristics		cART-Naïve	cART-Treated
*n*		147	317
Age	Years	44.1 (38.0–51.7)	49.2 (42.8–57.5)
Sex	Female	41 (27.9)	88 (27.8)
BMI	Kg/m^2^	18.5 (12.8–21.7)	23.9 (22.2–25.8)
Ancestry	European	119 (84.4)	216 (86.4)
	African	14 (9.9)	28 (11.2)
	South American	6 (4.3)	6 (2.4)
	Other	2 (1.4)	0
Risk Factors	Males who have sex with males	44 (33.6)	66 (28.1)
	Heterosexuals	41 (31.3)	60 (25.5)
	Intravenous drug users	24 (18.3)	80 (34.0)
	Other	6 (4.6)	6 (2.6)
	Unknown	15 (11.5)	23 (9.8)
Comorbidities	Liver cirrhosis	6 (4.2)	27 (10.9)
	Psychiatric disorders	25 (19.4)	46 (19.0)
	Past syphilis	30 (21.4)	54 (22.0)
	Chronic HCV	24 (16.8)	70 (28.2)
	Chronic HBV	14 (9.9)	32 (12.9)
	Toxoplasma positive serology	15 (31.2)	29 (42.0)
	Active smoking	45 (30.7)	89 (28.0)

BMI: Body mass index; HCV: Hepatitis C Virus; HBV: Hepatitis B Virus.

**Table 2 diagnostics-11-00867-t002:** Clinical and cART-related features of study participants.

Characteristics		ART-Naïve (*n* = 147)	ART-Treated (*n* = 317)
Diagnosis	Asymptomatic/control	8 (5.8)	64 (23.7)
	Primary infection	1 (0.7)	–
	Late presentation	77 (55.8)	17 (6.3)
	Other CNS disorders	11 (8.0)	44 (16.3)
	CNS opportunistic infections	19 (13.8)	34 (12.6)
	HIV-related encephalitis/escape/rebound	2 (1.4)	7 (2.6)
	HAND	14 (10.1)	77 (28.5)
	White matter hyperintensity	2 (1.4)	11 (4.1)
	Syphilis	1 (0.7)	13 (4.8)
	Neuro syphilis	3 (2.2)	3(1.1)
RMN	White matter hyperintensities	87 (64.4)	150 (68.8)
Therapy	PI	–	41 (12.9)
	INSTI	–	88 (28)
	MVC	–	16 (5)
	NNRTI	–	55 (17.4)
Number of drugs		–	3

CNS: Central nervous system; HAND: HIV-associated neurocognitive disorder; PI: Protease inhibitors; INSTI: Integrase strand transfer inhibitors; MVC: Maraviroc; NNRTI: Non-nucleoside reverse transcriptase inhibitors.

**Table 3 diagnostics-11-00867-t003:** Laboratory features and biomarkers according to treatment group. Variables were tested through Mann–Whitney (continuous variables) or Chi–square/Fisher’s exact test (binomial).

Characteristics	ART-Naïve (*n* = 147)	ART-Treated (*n* = 317)	*p* Values
Nadir CD4+ (cell/mm^3^)	49 (20–118)	97 (26.5–208)	0.002
CD4+ (cell/mm^3^)	57 (23–117)	366.5 (154.25–620.5)	<0.001
HIV DNA (copies/10^6^ PBMCs)	478.5 (138.25–2462.5)	123 (29–395)	0.002
Serum HIV RNA (log_10_ cps/mL)	5.42 (4.93–5.95)	1.28 (0.42–1.91)	<0.001
CSF HIV RNA (log_10_ cps/mL)	3.88 (3.07–4.7)	1.28 (0.04–2.08)	<0.001
HIV duration (months)	1.23 (0.43–125.63)	143.37 (44.98–225.96)	<0.001
Viral suppression (months)	0 (0–0)	18 (3.8–73)	<0.001
CSF cells	0 (0–3)	0 (0–2)	0.266
CSF proteins	49 (39–68.25)	48 (38–63)	0.557
CSF glucose	50 (45–57)	48 (51–63)	<0.001
Serum HIV RNA < 50 cps/mL	1 (0.7)	196 (70.8)	<0.001
CSF HIV RNA < 50 cps/mL	5 (3.6)	178 (63.3)	<0.001
NADIR CD4 < 200 cells/mm3	117 (84.2)	164 (71.3)	0.005
BBB impairment	52 (35.4)	72 (22.7)	0.005
CSAR	6.2 (4.3–8.7)	5.6 (4.1–7.6)	0.119
Intrathecal IgG Synthesis	0 (0–25)	0 (0–20)	0.280
t–Tau	165 (96–293)	222 (129–350)	0.045
p–Tau	33 (25–41)	37 (27–51)	0.040
Aβeta_42_	962 (704–1142)	919 (736–1171)	0.901
S100Beta	145 (48–260)	129 (70.25–219.5)	0.758
Neopterin	2.9 (1.4–5.7)	0.9 (0.5–1.9)	<0.001
JCV DNA+	8 (5.6)	17 (5.1)	1.000
EBV DNA+	32 (22.3)	32 (10.1)	0.001
CMV DNA+	8 (5.6)	5 (1.5)	0.030

VL: Viral load; CSF: Cerebrospinal fluid; PRO: Proteins; GLU: Glucose; BBB: Blood–brain barrier; CSAR: Cerebrospinal fluid Serum Albumin Ratio; ALB ind: Albumin index; p–Tau: Phosphorylated tau; Abeta1–42: beta-amyloid 1–42; S100B: S100beta; t–Tau: Total tau; JCV: JC virus; EBV: Epstein–Barr virus; CMV: Cytomegalovirus.

**Table 4 diagnostics-11-00867-t004:** Demographic and clinical features, plasma, and cerebrospinal fluid biomarkers in patients with normal and impaired blood–brain barrier. P values are calculated separately in naïve (first column) and treated (second column) participants.

Characteristics	Naïve Features			cART-Treated Features		
	Intact BBB	BBBi	*p* Values	Intact BBB	BBBi	*p* Values
*n*	95 (64.6)	52 (35.4)	–	245 (77.3)	72 (22.7)	–
Female sex	31 (32.6)	10 (19.2)	0.088	75 (30.6)	13 (18.1)	0.037
Age	46 (39–52)	41 (37–49)	0.034	49 (43–57)	48 (42–56)	0.854
CD4+ (cell/mm^3^)	47 (20–110)	72 (36–218)	0.132	371 (155–628)	357 (145–593)	0.910
Nadir CD4+ (cell/mm^3^)	46 (15–107)	71 (26–149)	0.089	95 (25–202)	100 (31–229)	0.283
Nadir CD4 < 200 cells/mm^3^	78 (87.6)	39 (78.0)	0.152	132 (73.3)	32 (64.0)	0.218
Serum HIV RNA (Log_10_ cps/mL)	5.4 (4.9–5.9)	5.6 (4.9–6.0)	0.098	<1.28 (<1.28–1.80)	<1.28 (<1.28–2.32)	0.598
Serum HIV RNA < 50 cps/mL	1 (1.1)	0 (0)	1.00	157 (72.7)	39 (63.9)	0.204
HIV DNA (copies/10^6^ PBMCs)	478 (76–2324)	903 (160–12064)	0.291	91 (<50–362)	147 (70–829)	0.160
HIV duration (months)	–	–	–	143 (45–226)	142 (42–225)	0.929
Viral suppression (months)	–	–	–	17 (2.5–65)	29 (9–95)	0.115
History of CNS opportunistic infections	6 (6.3)	13 (25)	0.002	18 (7.3)	16 (22.2)	0.001
Receiving NNRTIs	–	–	–	38 (15.5)	17 (23.6)	0.114
Receiving PIs	–	–	–	31 (12.7)	10 (13.9)	0.842
Receiving INSTIs	–	–	–	75 (30.9)	13 (18.3)	0.050
CSF cells ≥ 5/mm^3^	12 (13)	21 (41.2)	<0.001	20 (9.5)	17 (27.9)	0.001
CSF proteins	40 (34–49)	75 (56–90)	<0.001	43 (34–52)	81 (66–107)	<0.001
CSF glucose	51 (45–58)	48 (42–54)	0.022	57 (52–63)	56 (50–65)	0.940
CSF HIV RNA (Log_10_ cps/mL)	3.71 (2.82–4.39)	4.36 (3.49–5.09)	0.002	<1.28 (<1.28–2.03)	1.53 (<1.28–2.66)	0.029
CSF HIV RNA < 50 cps/mL	4 (4.3)	1 (2.1)	0.660	143 (65.3)	35 (56.5)	0.233
CSAR	5.0 (3.8–6.0)	9.6 (8.3–12.8)	<0.001	4.9 (3.8–6.2)	10.4 (8.8–13.4)	<0.001
Intrathecal IgG Synthesis (%)	38 (40.4)	15 (29.4)	0.210	76 (33.8)	17 (25.8)	0.234
tau	134 (93–306)	173 (136–285)	0.282	225 (134–344)	212 (113–403)	0.865
p–Tau	31 (24–41)	34 (27–42)	0.423	36 (26–51)	39 (29–50)	0.648
Aβeta_42_	925 (734–1144)	1017 (580–1149)	0.947	918 (726–1165)	950 (752–1284)	0.665
S100Beta	151 (55–281)	102 (38–229)	0.410	121 (71–202)	178 (60–351)	0.269
Neopterin	2.52 (1.19–4.80)	3.42 (2.10–9.80)	0.029	0.88 (0.50–1.70)	1.38 (0.50–3.40)	0.048
JCV DNA+	3 (3.2)	5 (9.6)	0.131	10 (4.1)	7 (9.7)	0.075
EBV DNA+	16 (16.8)	16 (30.8)	0.061	19 (7.8)	13 (18.1)	0.015
CMV DNA+	1 (1.1)	7 (13.5)	0.003	3 (1.2)	2 (2.8)	0.319

“BBB”, blood–brain barrier; “BBBi”, blood–brain barrier impairment; “CSF”, cerebrospinal fluid; “PI”, protease inhibitors; “INSTI”, Integrase strand transfer inhibitors; “NNRTI”, non-nucleoside reverse transcriptase inhibitors; “CSAR”, CSF to serum albumin ratio.

## Data Availability

The data presented in this study are available on request from the corresponding author.

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
