# Peer review of "Blood–Brain Barrier Impairment in Patients Living with HIV: Predictors and Associated Biomarkers"

_diagnostics, 2021, doi:10.3390/diagnostics11050867_

Round 1
Reviewer 1 Report
The manuscript entitled “Blood-brain Barrier Impairment in Patients Living with HIV: Predictors and Associated Biomarkers” by Caligaris et al. is an interesting paper describing and adding supporting new data to an important issue for the understanding the pathogenesis of neurocognitive disorders in people living with HIV (PLWH) and for identifying potential associated CSF/blood biomarkers. Indeed, as described, several grades of HIV-related cognitive impairment persist with effective ART and remain a clinical concern for PLWH, thus the identification of biomarkers with the potential for predicting the risk for neurocognitive impairment could be pivotal for the early diagnosis of HAND and the management of PLWH. The study aims at evaluating demographic, clinical and therapeutic characteristics, risk factors, comorbidities, as well as CSF and plasma biomarkers in both naïve and treated PLWH potentially associated with blood-brain barrier impairment (BBBi). The authors describe a cross-sectional study in PLWH including 464 participants (147 cART-naive and 317 on cART) and, by use of multivariate binary logistic regression, found multiple factors associated with BBBi and conclude that BBBi was prevalent in naïve and treated PLWH and was associated with CSF HIV RNA and neopterin. The article is generally clear and well written, the results are generally well presented and the conclusions drawn are supported by the results obtained, however the following points should be addressed to make this paper suitable for publication in Diagnostics.
Major points:
4.Discussion, Page 9, lines 192-196: regarding the sentence “Considering that, the current study shows three limitations worth mentioning: firstly, in the multivariate analysis not all of these factors have been considered as they were not available; secondly, a wider range of patients was treated with INSTI compared to those treated with other classes of drugs (28%, versus 17.4% NNRTI treated patients, 12.9% treated with PI and 5% with Maraviroc).”, the third limitation is lacking. The authors should please rephrase or complete the sentence, accordingly.
The authors should please briefly comment (in the Discussion section) on the significance of the other CSF biomarkers evaluated in the naive versus cART population, in particular tau, p-tau, S100b, and Ab1–42, which results are described in the Results section (lines 107 to 109) and reported in table 3.
Minor points:
Abstract, Page 1, line 13: the authors should please spell out the acronym CNS
Abstract, Page 1, line 14: the authors should please spell out the acronym CSF
1.Background, Page 1, line 31: the authors should please spell out the acronym CNS
1.Background, Page 2, line 46: according to this reviewer the reference [10] after the sentence “beta-amyloid and Tau abnormalities [10]” does not seem correct, please check.
2.Materials and Methods, Page 2, line 66: the sentence “(normal if below 6.5 in patients aged <40 years and below 8 in patients 40 years) should be changed to “normal if below 6.5 in patients aged <40 years and below 8 in patients >40 years”
3.Results, Page 3, line 101: the sentence “others CNS disorders” should be changed to “other CNS disorders”
3.Results, Page 4, Table 2: in the present table “HAND” is mentioned twice with different percentages, the first time in the section “Diagnosis” of table 2, the second time after the section “Neurocognition”, which one is correct and what the percentages mentioned the second time refer to?
3.Results, Page 5, Table 3: the authors should please correct the formatting errors related to the number of patients cART-naïve (n=147) and cART-treated (n=317)
3.Results, Page 6, Table 4: the authors should please correct the formatting errors related to the values of CSF HIV RNA (Log10 cps/mL)
4.Discussion, Page 8, line 156: according to this reviewer the reference [15] after the sentence “progression in the most severe forms of chronic liver diseases.[15]” does not seem correct, please check.
4.Discussion, Page 9, lines 162-164: the sentence “Some authors, as Maki PM and Martin-Thormeyer, highlight that could be a higher risk of cognitive disorder for HIV-infected women than HIV-infected men. [18]” should be changed to “Some authors, as Maki and Martin-Thormeyer, highlight that there could be a higher risk of cognitive disorder for HIV-infected women than for HIV-infected men. [18]”
4.Discussion, Page 9, lines 166-167: the sentence “By contrast, a recent study who investigated prevalence and risk factors of HAND in Uganda show a higher risk for men compared with women.[20]” should be changed to “By contrast, a recent study who investigated prevalence and risk factors of HAND in Uganda shows a higher risk for men compared with women.[20]”
4.Discussion, Page 9, line 180: the authors should please spell out the acronym NFL
4.Discussion, Page 9, line 192: according to this reviewer the reference [27] in the sentence “factors, e.g. age and ethnicity. [27]” does not seem correct. Did the authors instead mean reference 26? Reference [26] is lacking in the manuscript. Please check.
4.Discussion, Page 9, lines 197-200: accordingly to the previous point, references [27] and [28] in the sentence “The association found between BBBi and CSF viral load, both among ART-naïve and treated patients, confirms the evidence observing an increase in CSF HIV-RNA among patients affected by neurological symptoms [28] which was associated with BBBi. [27]” should be rechecked and renumbered if necessary.
4.Discussion, Pages 9-10: References 30 to 32 are not reported in the text. The authors should please check and renumber the references accordingly.
4.Discussion, Page 10, lines 214-218: the sentence “In this regard, in pre-cART era CMV coinfection caused a major risk of progression to AIDS, several studies show that even in cART epoch the presence CMV in the blood is associated with a worse prognosis, cause of the increased risk of CMV disease progression, AIDS defining diagnosis and death.[35” should be changed to “In this regard, as in pre-cART era CMV coinfection caused a major risk of progression to AIDS, several studies show that even in cART epoch the presence of CMV in the blood is associated with a worse prognosis, cause of the increased risk of CMV disease progression, AIDS-defining diagnosis and death.[35]”
4.Discussion, Page 10, lines 218-220: the sentence “As reported in literature, our results confirm the less prevalence of CMV infection in cART treated patients, although CMV rest a relevant comorbidity even in cART ages.” should be changed to “As reported in literature, our results confirm the less prevalence of CMV infection in cART treated patients, although CMV remains a relevant comorbidity even in the cART era.”
4.Discussion, Page 10, lines 223-226: the meaning of the sentence “In this context, even though a French study found a similar rate of EBV detection among HIV-infected subjects and controls, [37] our results sustain the evidence that described EBV in peripheral blood higher in PLWH than non-infected subjects and they support the usefulness of evaluating EBV DNA in HIV+ subjects. [38]” is not clear to this reviewer, please reframe.
4.Discussion, Page 10, lines 228-229: the sentence “HAND could be a direct consequence of the BBB breakdown by different mechanism, such as an increased permeability, that allow an increasing entry…” should be changed to “HAND could be a direct consequence of the BBB breakdown by different mechanisms, such as an increased permeability, that allow an increasing entry…”
5.Conclusions, Page 10, line 238: the word “astrocysosis” should be changed to “astrocytosis”
Bibliography, Pages 10-12: accordingly to the instruction for Authors, references should be reported as follows: 1. Author 1, A.B.; Author 2, C.D. Title of the article. Abbreviated Journal Name Year, Volume, page range. The authors should please check that all references are complete and written in English.
Other points:
Accordingly to the instruction for Authors, in the text reference numbers should be placed in square brackets [ ], and placed before the punctuation; for example [1], [1–3] or [1,3]. It seems to this reviewer that the reference numbers are placed in the text after the punctuation. Please check.
Author Response
Thank you for your comments and suggestions,
Please see the attachments for the point-by-point response and the new version of the article.

Reviewer 2 Report
This is a cross-sectional study investigating blood brain barrier impairment (BBBi) in 464 people living with HIV (PLWH). Using cerebrospinal fluid (CSF) collected from infected individuals who were either untreated or were on ART, the authors measured multiple clinical features including plasma and CSF values, using CSF to serum albumin ratio as an indicator of BBBi. In addition, demographic information, risk factors and comorbidities of the study participants were provided. Statistical analyses of the different characteristics revealed a potential connection between several risk / protective factors and BBBi in PLWH. The analyses suggest that BBBi might still be prevalent even in the presence of cART, although to a lesser extent than untreated individuals. Being male, having a history of CNS opportunistic infections, CSF HIV RNA, and nadir CD4 cell counts were found to be independently associated with BBBi.
It’s a study with a large number of participants, and the information is useful to the medical/virological community. I find that the results could have been presented in a somewhat clearer way. The way it stands now, it is rather difficult to grasp the few key take away points from the results section.
It is also not entirely clear why this study finds connections that other studies did not find, or whether there were differences in the analytical/statistical methods used. These should be included in the discussion section in a clearer way. Sometimes other studies are referred to by author names (Maki PM and Martin-Thormeyer), sometimes they are referred to only by nationality (the French or Ugandan study). These references should be kept consistent.
Gender and sex are not synonymous. When describing the anatomical/biological concept, the word sex should be used. Gender, on the other hand, is a social construct and its use is not accurate here. (Table 4, Figure 1 legend, also lines 120, 154, 155 among many others.)
Some other minor issues:
The language in discussion could use some minor editing. E.g. I’m not sure what is meant by this sentence: “..despite several studies tried to identify the treatment with the more effectiveness on CNS..”
There are many details that need to be corrected, such as punctuation marks, spaces, paragraph indents, superscripts/subscripts etc. which I find are beyond the reviewers’ responsibilities.
E.g. in Table 4: HIV DNA (copies*106 PBMCs)
Bibliography 1st item: “consultato” should be translated to English.
L187: “ed” meaning and?
When the word “univocal” is used I wonder if the authors mean unequivocal? (lines 162, 183)
L219: “CMV rest” -- do they mean CMV remains?
Author Response

(The authors gave the same response as above.)

Round 2
Reviewer 1 Report
The authors adequately responded to all comments raised by this referee, thus the paper appears to be suitable for publication in Diagnostics.